# A more accurate method for colocalisation analysis allowing for multiple causal variants

**Chris Wallace**[1,2]*

**1** Cambridge Institute of Therapeutic Immunology and Infectious Disease, University of Cambridge, Cambridge, United Kingdom, **2** MRC Biostatistics Unit, University of Cambridge, Cambridge, United Kingdom

* cew54@cam.ac.uk

**Data Availability Statement:** Code to perform the simulations may be found at https://github.com/chr1swallace/coloc-susie-paper. A version of coloc including SuSiE is available from CRAN at https://cran.r-project.org/package=coloc.

## Abstract

In genome-wide association studies (GWAS) it is now common to search for, and find, multiple causal variants located in close proximity. It has also become standard to ask whether different traits share the same causal variants, but one of the popular methods to answer this question, coloc, makes the simplifying assumption that only a single causal variant exists for any given trait in any genomic region. Here, we examine the potential of the recently proposed Sum of Single Effects (SuSiE) regression framework, which can be used for fine-mapping genetic signals, for use with coloc. SuSiE is a novel approach that allows evidence for association at multiple causal variants to be evaluated simultaneously, whilst separating the statistical support for each variant conditional on the causal signal being considered. We show this results in more accurate coloc inference than other proposals to adapt coloc for multiple causal variants based on conditioning. We therefore recommend that coloc be used in combination with SuSiE to optimise accuracy of colocalisation analyses when multiple causal variants exist.

## Author summary

Genetic association studies have found evidence that human disease risk or other traits are under the influence of genetic variants. As results of studies are made publicly available, more research focuses on whether different traits are under influence of the same variants, which may help us understand how variants lead to differences in disease risk. However, one of the popular methods to answer this question, coloc, makes the simplifying assumption that no two members of the set of causal variants for any one trait are close to each other. Here, we examine the potential of the recently proposed Sum of Single Effects (SuSiE) regression framework, for use with coloc. SuSiE is a novel approach that allows evidence for association at multiple causal variants in proximity to be evaluated simultaneously. We show this results in more accurate coloc inference than other proposals to adapt coloc for multiple causal variants based on conditioning. We therefore recommend that coloc be used in combination with SuSiE to optimise accuracy of colocalisation analyses when multiple causal variants exist.

**Funding:** CW is funded by the Wellcome Trust (WT107881, WT220788) and the MRC (MC UU 00002/4). This study was also supported by the NIHR Cambridge BRC (BRC-1215-20014). The views expressed are those of the author(s) and not necessarily those of the NHS, the NIHR or the Department of Health and Social Care. The funders had no role in study design, data collection and analysis, decision to publish, or preparation of the manuscript.

**Competing interests:** The authors have declared that no competing interests exist.

This is a *PLOS Genetics* Methods paper.

# Introduction

Colocalisation is a technique used for assessing whether two traits share a causal variant in a region of the genome, typically limited by linkage disequilibrium (LD). In its original form, it made the simplifying assumption that the region harboured at most one causal variant per trait [1], and we begin by explaining how that enables inference to be made quickly, using only GWAS summary statistics, and without information about LD. The approach begins by enumerating all *variant-level hypotheses*—the possible pairs of causal variants (or none) for the two traits—and the relative support for each in terms of Bayes factors, calculated from GWAS effect estimates at each SNP and their standard errors [2]. Thanks to the single causal variant assumption, each one of these combinations is associated to exactly one *global hypothesis*

$H_0$: no association with either trait in the region

$H_1$: association with trait 1 only

$H_2$: association with trait 2 only

$H_3$: both traits are associated, but have different single causal variants

$H_4$: both traits are associated and share the same single causal variant

The second step calculates log Bayes factors for each of these global hypotheses by summing the log Bayes factors for all corresponding variant-level hypotheses. Finally, standard combination of Bayes factors with prior probabilities of each hypothesis allows us to calculate posterior probabilities. A full exposition of these steps are found in [3]. Note that the per-SNP Bayes factors relate closely to fine mapping, because they are proportional to fine mapping posterior probabilities of causality under a single causal variant assumption [4]. Thus we can calculate fine mapping posterior probabilities from the single trait Bayes factors, or from the coloc Bayes factors if we are sufficiently convinced of $H_4$ to produce probabilities that combine information from both traits.

The single causal variant assumption implies that each pair of variants being causal for the two traits are mutually exclusive events. However, the assumption is unrealistic, as multiple causal variants may exist in proximity, which also challenges the definition of colocalisation as presented above as none of the global hypotheses encompass multiple causal variants. Alternative methods for colocalisation have been developed which do not make this assumption. eCAVIAR [5] uses the CAVIAR [6] approach (which accommodates multiple causal variants) to fine map each trait, and gives probabilities that any variant is causal for both traits as the product of the single trait causal probabilities. However, this treats causality at each trait as independent events, when there is abundant evidence that a SNP causal for one trait is more likely to be causal for another. Alternatively, HEIDI/SMR [7] uses a frequentist framework, treating the null hypothesis as colocalisation, and rejecting this when there is evidence against. Here, multiple causal variants are dealt with by requiring colocalisation across all causal variants in a region, and that the effects of each causal variant on the two traits is proportional. That is, if one causal variant has a two-fold greater effect on trait 1 compared to trait 2, then all other causal variants are assumed to also have a two-fold greater effect.

Unlike these, coloc works with a single pair of causal variants at a time, and explicitly allows incorporating any expectation that causal variant are likely to be shared through prior probabilities. In previous work, [3] we allowed for multiple causal variants in coloc by using conditional regression to distinguish lead variants, with the added requirement of supplying an LD

matrix for the variants under test. Each pair of lead variants could be examined by a single coloc run, leading to multiple colocalisation comparisons. Thus, if trait 1 had two causal variants tagged by SNPs A and B and trait 2 had one, tagged by SNP C, we would conduct two colocalisation analyses, to ask whether A and C corresponded to a shared causal variant, and whether B and C corresponded to a shared causal variant. This allows the simple combination of log Bayes factors through summation, but explicitly assumes that data can be decomposed into layers corresponding to the causally distinct signals. The stepwise regression approach upon which conditioning is based is known generally to produce potentially unreliable results [8], a phenomenon that can be exacerbated by the extensive correlation between genetic variants caused by LD [9]. Thus, this solution remains unsatisfactory.

A suite of Bayesian fine-mapping methods have been developed recently which calculate posterior probabilities of sets of causal variants for a given trait [6, 10, 11]. However, the marginal posterior probabilities calculated from these are no longer mutually exclusive events, so they could not be easily adapted to the colocalisation framework. An alternative would be to consider all possible combinations of models between two traits, but this combinatorial problem is computationally expensive [9]. Recently, the Sum of Single Effects (SuSiE) regression framework [12] was developed which reformulates the multivariate regression and variable selection problem as the sum of individual regressions each representing one causal variant of unknown identity. This allows the distinct signals in a region to be estimated simultaneously, and enables quantification of the strength of evidence for each variant being responsible for that signal. Conditional on the regression being considered, the variant-level hypotheses are again mutually exclusive. Here we describe the adaptation of coloc, allowing for multiple labelled comparisons in a region, to use the SuSiE framework and demonstrate improved efficacy over the previously proposed approaches. While SuSiE is written in terms of the full genotype matrix, it has been extended to require only summary statistics by combination with a "regression with summary statistics" likelihood formulation [13]. We use the summary statistic module of SuSiE, `susie_rss()`, so that the format of data currently expected by coloc, GWAS summary statistics for each trait and an LD matrix, is unchanged.

## Methods

### Adaptation of coloc approach

The new `coloc.susie()` function in the coloc package (https://github.com/chr1swallace/coloc/tree/susie) takes a pair of summary datasets in the form expected by other coloc functions, runs SuSiE on each and performs colocalisation as described below. We use the `susie_rss()` function in the susieR package to fine-map each summary statistic dataset, run with default options, although the `susie.args` argument in `coloc.susie()` allows arguments to be supplied to `susie_rss()`. `susie_rss()` returns a matrix of variant-level Bayes factors for each modelled signal and a list of signals for which a 95% credible set could be formed, corresponding to a subset of rows in the matrix of Bayes factors. These rows are then analysed in the standard coloc approach, for every pair of regressions with a detectable signal across traits. Explicitly, if $L_1$ and $L_2$ signals are detected (have a credible set returned) for traits 1 and 2 respectively, then the colocalisation algorithm is run $L_1 \times L_2$ times. Thus, the user is presented with two lists of signals for each trait, and the $L_1 \times L_2$ matrix of pairwise posterior probabilities of $H_4$ may be examined to infer which pair of tags, if any, represent the same signal.

### Simulation strategy

We examined the performance of using SuSiE with coloc by simulation. We downloaded haplotypes for EUR samples in the 1000 Genomes phase 3 data [14], phased by IMPUTE2 [15],

from https://mathgen.stats.ox.ac.uk/impute/1000GP_Phase3.html. We used lddetect [16] to divide the genome into approximately LD-independent blocks, and extracted haplotypes consisting of 1000 contiguous SNPs with MAF > 0.01. We simulated case-control GWAS summary statistics for a study with 10,000 cases and 10,000 controls, corresponding to the LD and MAF calculated from these haplotypes using simGWAS [17], with one or two common causal variants (MAF > 0.05) chosen at random and log odds ratios sampled from $N(0, 0.2^2)$. We discarded any datasets which did not have a minimum $p < 10^{-6}$ to match our expectation that fine-mapping and colocalisation are only conducted when there is at least a nominal signal of association. We simulated 100 such datasets for each of 100 randomly selected LD blocks, and sampled from these sets of summary data for all the simulations detailed below.

We repeatedly simulated GWAS summary data for a single trait with one or two causal variants in small or large genomic regions (1000 or 3000 SNPs, where 3000 SNP regions were constructed by concatenating three 1000 SNP datasets). We constructed pairs of simulated data for two traits, such that each trait had one or two causal variants and each pair of traits shared zero, one or two causal variants. We simulated 10,000 examples from each collection, with each example analysed independently. Analysis compared different approaches:

1. **single** single causal variant coloc analysis of every pair of traits

2. **cond_it** multiple causal variant coloc analysis using a conditioning approach to allow for multiple causal variants, iterative mode

3. **cond_abo** multiple causal variant coloc analysis using a conditioning approach to allow for multiple causal variants, "all but one" mode

4. **susie** multiple causal variant coloc analysis using SuSiE to allow for multiple causal variants

Conditioning can be run in two modes. Assume that stepwise regression detects two signals, tagged by SNPs A and B. In the iterative mode, we first use the raw data in a first step, and then the data conditioned on A in a second step. This corresponds to how stepwise identification of independent signals in GWAS is commonly approached. An alternative is to condition on B in the first step, and A in the second step, attempting to isolate the separate signals. This corresponds more closely to the hope in multiple causal variant coloc that we can decompose the data into layers corresponding to the separate signals. However, because the identification of the second signal B is likely to be more uncertain than A (because it is weaker, and was detected through conditioning on the already uncertain A), it may introduce further error.

In order to assess the accuracy of each coloc analysis, we needed to assess whether the comparison corresponded to a case of shared or distinct causal variants. For each signal passed to coloc, we identified the variant with the highest posterior probability of causality, $v_1$ and $v_2$ for traits 1 and 2 respectively (it is possible that $v_1 = v_2$). We then labelled the variant $v_i$ ($i = 1, 2$) according to the rules:

**A** $r^2(v_i, A) > 0.5 \wedge r^2(v_i, A) > r^2(v_i, B)$

**B** $r^2(v_i, B) > 0.5 \wedge r^2(v_i, B) > r^2(v_i, A)$

- otherwise

If either of the variants was labelled "-" then the comparison was labelled "unknown". Otherwise it was labelled by the concatenation of the two labels. We compared the average posterior probability profiles between methods, stratified according to this labelling scheme.

Results in this manuscript were generated using R version 4.0.4 with packages susieR version 0.11.42 and coloc version 5.1.0.

## Results

Summary results of the coloc simulation study are given in S1 Table, and presented graphically in Fig 1 and S1 Fig. When when both traits really did contain only a single causal variant, we found that single coloc generally performed best (top two rows of Fig 1). SuSiE-based analysis appeared to lose a little power (lower bar heights indicating fewer comparisons performed) but was equally accurate amongst comparisons performed. The situations when coloc-SuSiE did not perform any comparisons corresponded to cases where SuSiE did not identify any

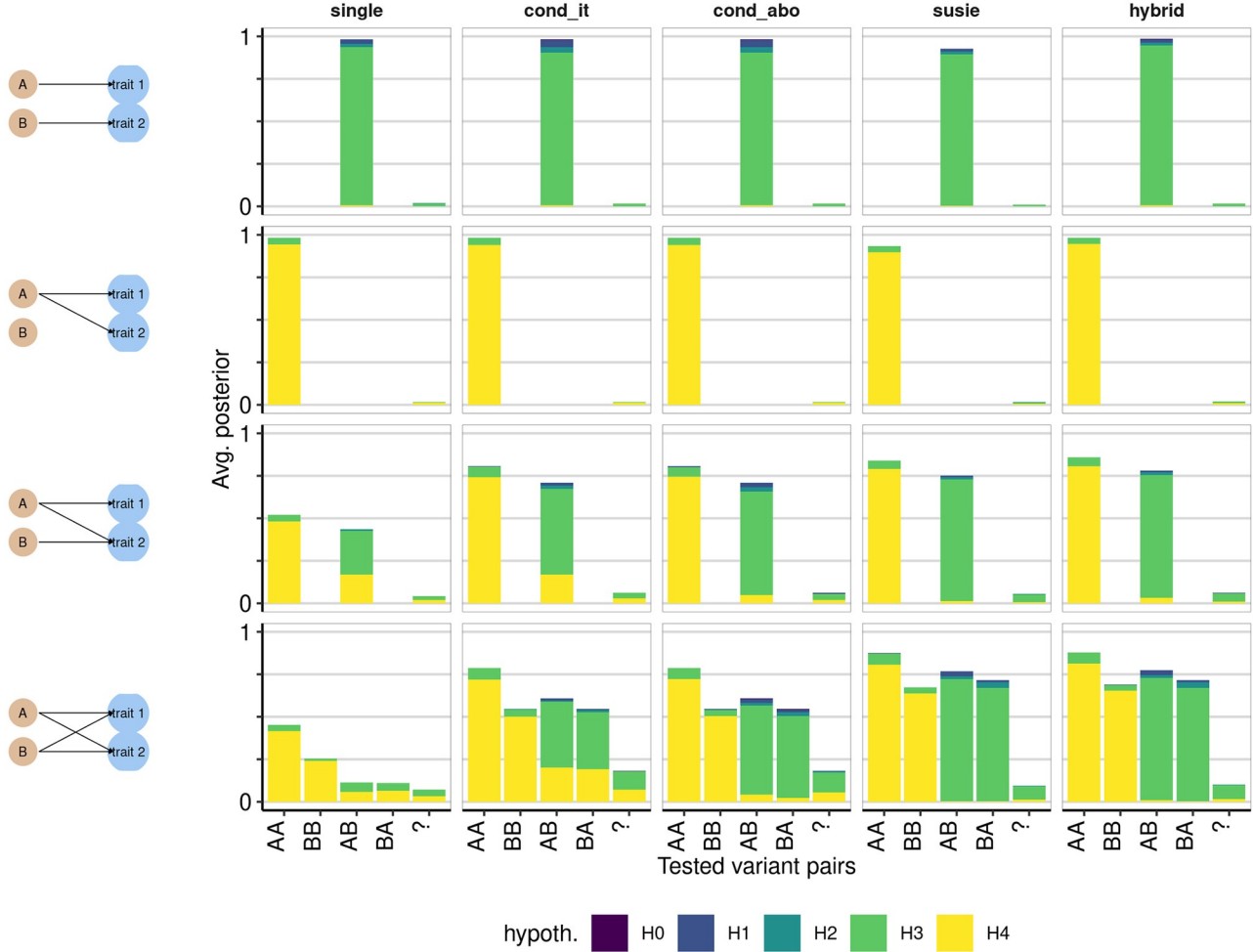

**Fig 1. Average posterior probability distributions in simulated data.** The four classes of simulated datasets are shown in four rows, with the scenario indicated in the left hand column. For example, the top row shows a scenario where traits 1 and 2 have distinct causal variants A and B. Columns indicate the different analysis methods, with susie indicating SuSiE, cond_it indicating that coloc-conditioning was run in iterative mode, and cond_abo indicating it was run in "all but one" mode. For each simulation, the number of tests performed is at most 1 for "single", or equal to the product of the number of signals detected for the other methods. For each test, we estimated which pair of variants were being tested according to the LD between the variant with highest fine-mapping posterior probability of causality for each trait and the true causal variants A and B. If $r^2 > 0.5$ between the fine-mapped variant and true causal variant A, and $r^2$ with A was higher than $r^2$ with B, we labeled the test variant A, and vice versa for B. Where at least one test variant could not be unambigously assigned, we labelled the pair "?". The total height of each bar represents the proportion of comparisons that were run for that variant pair, out of the number of simulations run, and typically does not reach 1 because there is not always power to perform all possible tests. Note that because we do not limit the number of tests, the height of the bar has the potential to exceed 1, but did not do so in practice. The shaded proportion of each bar corresponds to the average posterior for the indicated hypothesis, defined as the ratio of the sum of posterior probabilities for that hypothesis to the number of simulations performed. Recall that $H_0$ indicates no associated variants for either trait, $H_1$ and $H_2$ a single causal variant for traits 1 and 2 respectively, $H_3$ and $H_4$ that both traits are associated with either distinct or shared causal variants, respectively. Each simulated region contains 1000 SNPs.

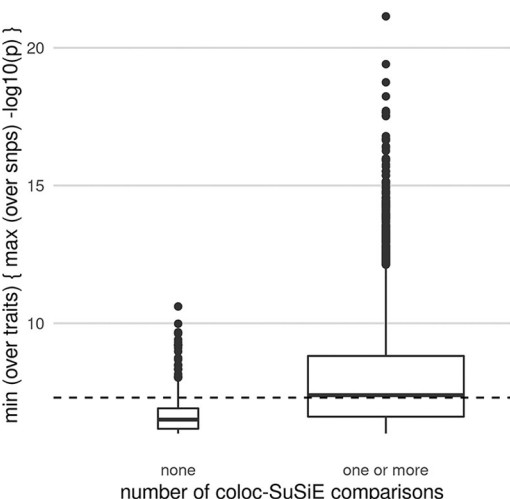

**Fig 2. Distribution of maximum -log10 p values for simulated datasets where coloc-SuSiE could find at least one credible set for each trait, or could not.** Each dataset was summarised by its maximum -log10 p value, and the pair of datasets by the minimum of these. A dashed line shows the conventional GWAS significance threshold of $5 \times 10^{-8}$. This shows that when coloc-SuSiE does not produce any results it is generally in cases of lower power.

credible sets for one or both traits, which were likely to be examples with higher minimum p values (Fig 2). A hybrid approach, running coloc-SuSiE if possible, and coloc-single if not outperformed any other strategy. When either one or both traits had two causal variants (bottom two rows of Fig 1), SuSiE outperformed all other methods in terms of accurately calling "AB" comparisons distinct ($H_3$) rather than shared ($H_4$) and performed as many or more comparisons than the other coloc methods. Hybrid SuSiE-single-coloc was very similar to SuSiE-coloc, or marginally better. In the two causal variant cases, single coloc tended to equivocate between $H_3$ and $H_4$ when testing AB-like signals in the presence of a shared causal variant (ie where the peak signals in each trait related to distinct causal variants) which should be inferred $H_3$. This relates to a known feature of coloc, which may detect the colocalising signal even when additional non-colocalising signals are present [1].

This feature also presents problems for the conditioning approach cond_it, as demonstrated by the high average posterior probability for $H_4$ in the "AB" comparisons, one of which is examined in detail in Fig 3. In this example, trait 1 has one causal variant, A, whilst trait 2 has two, A and B, with B having slightly greater significance. In the first round of analysis by the conditioning method, the original sets of summary statistics are passed to coloc. Because A is the stronger effect for trait 1, the test is labelled "AB", but gives a high posterior to $H_4$ because there is one shared causal variant (A). Then the stronger effect, B, is conditioned out, and the analysis rerun with trait 1, and trait 2 conditioned on B. This test again gives a high posterior for $H_4$. This situation is confusing, because the same signal in trait 1 appears to colocalise with different signals in trait 2. SuSiE models both signals simultaneously, so we can attempt to colocalise trait 1 with each signal independently, finding high $H_3$ for one and high $H_4$ for the other. If we were confident we could infer both the exact number of independent signals and their identity correctly by conditioning, we could attempt to emulate this in the conditioning, using the "all but one" rather than "iterative" mode. This does result in better average performance than the iterative mode (Fig 1). However it is often outperformed by SuSiE. S2 Fig shows an example where the stepwise approach cond_abo is less able to correctly

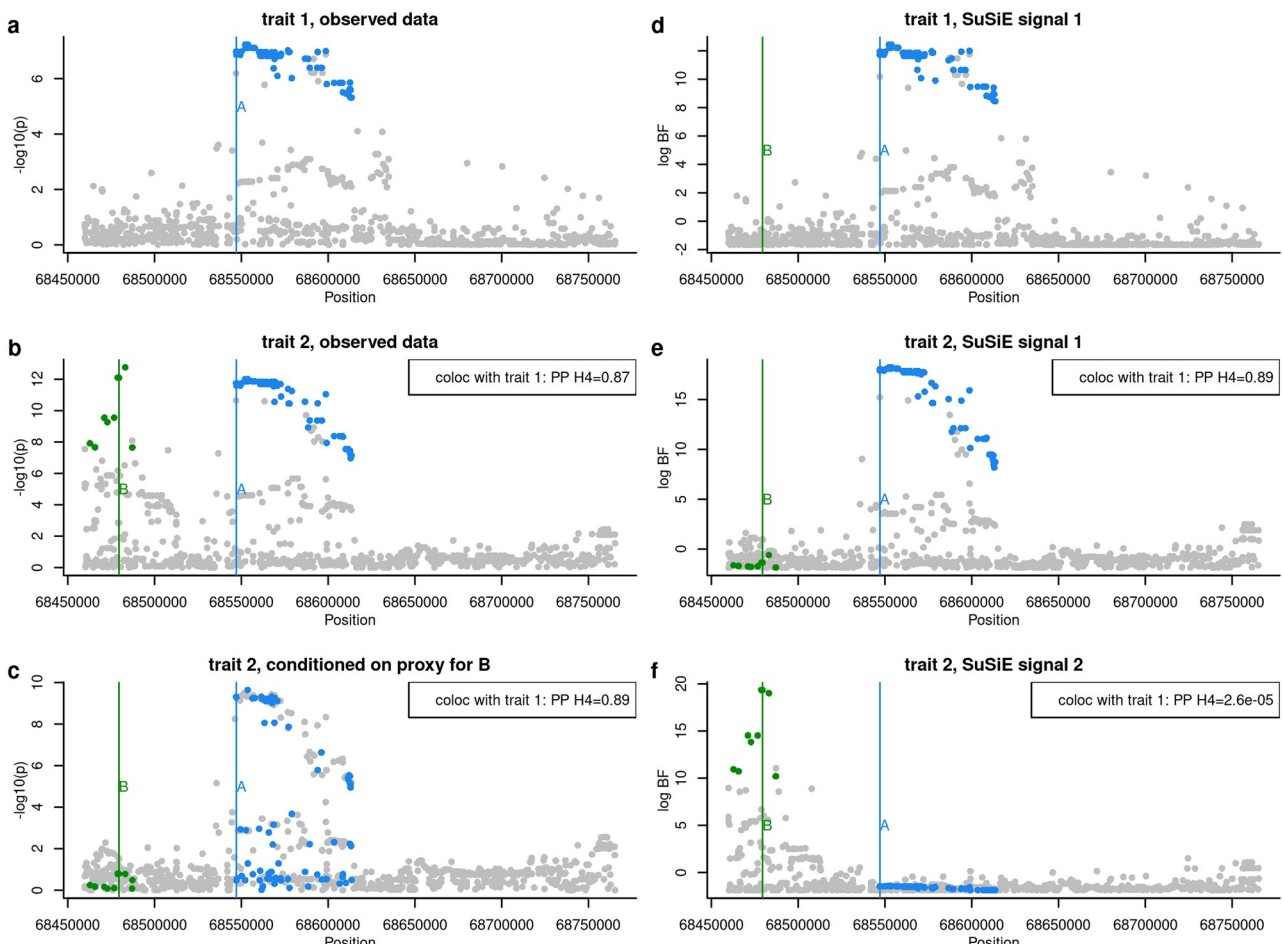

**Fig 3. Example where the conditional coloc approach, run in iterative mode, finds misleading results. a** and **b** show the "observed" data (simulated from 1000 SNPs with MAF > 0.01) as -log₁₀ p values for traits 1 and 2 respectively. Trait 1 has one causal variant, A, and trait 2 has two, A and B. Conditioning identifies a second independent signal for trait 2, and the results of conditioning on the strongest signal is shown in **c**. Coloc comparisons are based on $(a, b)$ and $(a, c)$ and both find the posterior probability (PP) of the shared causal variant hypothesis $H_4$ is > 0.8. SuSiE analysis of the same data finds one credible set in trait 1, and log₁₀ Bayes factors (BF) for this are shown in **d**. It finds two credible sets for trait 2, and the log₁₀ BF for these are shown in **e** and **f**. Coloc comparisons are based on $(d, e)$ and $(d, f)$ and find PP of $H_4$ of > 0.9 and $< 10^{-4}$ respectively. Blue and green points are used to highlight SNPs in LD with ($r^2 > 0.8$) the true causal variants A and B respectively. The data underlying this figure are available in S1 Data.

identify the separate signals. The A signal is not well identified, and therefore not be adequately conditioned out, which may results in two apparently different comparisons with trait 1 which both produce a high $H_4$. In this example too, SuSiE more correctly produces two comparisons, one with high $H_3$ and one with high $H_4$.

Finally, we compared the different approaches in terms of their ability to pinpoint the causal variant by SNP-level posterior probabilities of causality conditional on colocalisation. This vector of posterior probabilities is returned as a side-effect of every coloc comparison, and we would expect the posterior probability at the causal variant to increase when colocalisation ($H_4$) is called correctly. We took all simulation results which gave $P(H_4|\text{Data}) > 0.9$, and examined the distribution posterior probabilities at the causal variant (Fig 4). We found the expected pattern for single and SuSiE based coloc, but conditioning did not generally result in a higher posterior probability, presumably because the difficulty with these approaches such as that exemplified in Fig 3.

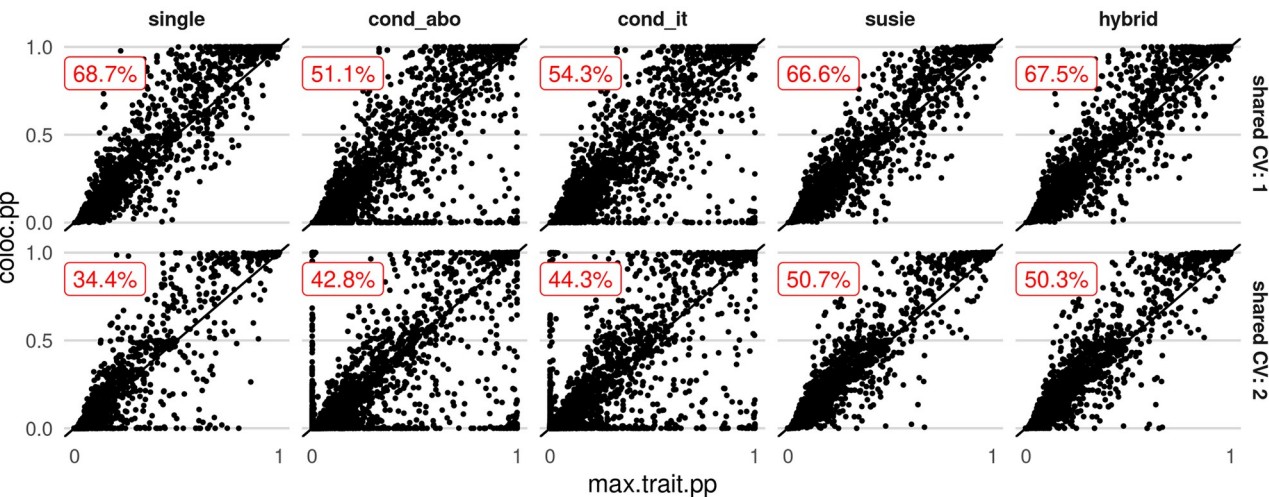

**Fig 4. Fine mapping posterior probabilities at causal variants in single trait and coloc analysis, amongst datasets with high probability of colocalisation ($P(H_4|\text{Data}) > 0.9$) according to the method shown.** Each point represents one causal variant in a dataset; its x location shows its maximum fine mapping posterior probability (PP) in either single trait, its y location shows its PP after coloc. Results are divided by rows into those from datasets with 1 (top) or 2 (bottom) causal variants, and by columns according to method. The text in red shows the percent of datasets which led to an increase in PP at causal variants after coloc analysis.

## Discussion

While coloc has been a popular method for identifying sharing of causal variants between traits, the common simplifying assumption of a single causal variant has been criticised, because it does not accord with findings that causal variants for the same trait may cluster in location (e.g. because they act via the same gene) [18]. Using the new SuSiE framework to partition the problem into multiple coloc comparisons and assuming the single causal variant assumption holds in each appears to resolve this issue better than the previously proposed conditional approach. It allows multiple signals to be distinguished, and then colocalisation analysis conducted on all possible pairs of signals between the traits. However, when no credible sets can be detected with confidence by SuSiE, single-coloc may still be able to make some inference. This can improve power when there really is one causal variant per trait, but doesn't appear to cause incorrect inference in the low powered multiple-causal cases. Thus we recommend a hybrid approach be adopted, using coloc-SuSiE where possible, but falling back on coloc-single when SuSiE cannot identify any credible sets.

Note that in earlier preprints of this manuscript, we suggested an approach based on trimming input data to decrease the computational time required to run `susie_rss`, but more recent versions of susieR, including the one used here, are faster and so we no longer consider that approach to be required.

This manuscript presents one approach to colocalisation in the case of multiple causal variants, that assumes that distinct signals can be decomposed even if physically proximal, which SuSiE appears to do admirably well. This framing of the colocalisation problem implicitly assumes there are a finite number of causal variants for any trait which can be identified, and that traits may be compared in terms of their causal variants to identify shared variants. However, the concept of regional colocalisation can be approached in other ways in the multiple causal variant scenario. One approach reduces the possible hypotheses to two, with the alternative hypothesis corresponding to the existence of a causal variant in a region shared by two (or

more) traits. [19] Another focuses on a variant-level definition of colocalisation, estimating the probability that each variant in turn is causal for two traits, whilst allowing that other causal variants (shared or non-shared) may exist in the vicinity [5]. In contrast, the approach proposed here allows the number hypotheses tested to be determined by the data: it is the product of the number of credible sets identified by SuSiE for each trait. Whilst it relaxes the assumption of a single causal variant, one obvious caveat is that we have not yet reached (nor may we ever reach) sample sizes which enable all causal variants to be identified. Missed causal variants will provide incomplete comparisons of traits. It is also established that in lower power situations, even Bayesian fine-mapping methods that simultaneously model causal variants may identify a single SNP which tags two or more causal variants [9] and the interpretation of non-colocalisation at such false signals is likely to be misleading. On the other hand, it does seem useful to go beyond asking whether at least one causal variant is shared, and the attempt to both isolate and count the distinct causal variants per trait may be useful in designing follow-up experiments. As we better understand the architecture of complex traits, and design methods that accomodate the multiple causal variants that have been discovered, it is important to bear in mind that results will continue to be limited by sample size, and limited ability to detect rarer variants or those in regions of particular allelic heterogeneity, which even sophisticated methods such as SuSiE may find challenging.

## Supporting information

**S1 Table. Results of colocalisation simulations.** The columns shown are: scenario: the simulated causal variants in traits 1 and 2, for example A-AB indicates trait 1 has causal variant A and trait 2 has causal variants A and B. nsnps_in_region: Number of SNPs in simulated region (1000, 3000). method: method used for coloc analysis inferred_cv_pair estimated pair of causal variants under test. H0,H1,H2,H3,H4 average posterior support for each hypothesis. This is calculated as the sum of posterior probabilities for each hypothesis / number of simulations run. As some variant pairs are unlikely to be tested (eg the pair AA is unlikely to be tested in the scenario A-B) this is not the expected posterior support given AA is tested.
(CSV)

**S1 Fig. Companion to Fig 1, showing the results for simulated datasets with 3000 SNPs.** Legend otherwise as for Fig 1.
(TIF)

**S2 Fig. Example where the conditional coloc approach, run in "all but one" mode finds misleading results. a** and **b** show the observed data (-$\log_{10}$ p values) for traits 1 and 2 respectively. Conditioning identifies two independent signals for trait 2, and the results of conditioning on the signal closest to causal variants A and B are shown in **c** and **d** respectively. Coloc comparisons are based on (*a*, *c*) and then (*a*, *d*). SuSiE analysis of the same data finds one signal in trait 1, and $\log_{10}$ Bayes factors (BF) for this signal are shown in **e**. It finds two signals for trait 2, and the $\log_{10}$ BF for these are shown in **f** and **g**. Coloc comparisons are based on (*e*, *f*) and (*e*, *g*). The boxes on the lower plots show the results of running coloc analysis on that dataset against the data for trait 1 shown in **a** or **e** as appropriate. The data underlying this figure are available in S1 Data.
(TIF)

**S1 Data. Datasets plotted in Figs 4 and S2, including summary statistics and the underlying LD and MAF.**
(ZIP)

## Acknowledgments

We thank Stasia Grinberg and Anna Hutchinson for comments on an earlier version of this manuscript, and Matthew Stephens for detailed explanation of the computational complexities in the `susie_rss` function.

## Author Contributions

**Conceptualization:** Chris Wallace.

**Formal analysis:** Chris Wallace.

**Investigation:** Chris Wallace.

**Methodology:** Chris Wallace.

**Project administration:** Chris Wallace.

**Software:** Chris Wallace.

**Writing – original draft:** Chris Wallace.

**Writing – review & editing:** Chris Wallace.

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
