## [Decision Letter · Decision Letter 0]

29 Apr 2021

Dear Dr Wallace,

Thank you very much for submitting your Research Article entitled 'A more accurate method for colocalisation analysis allowing for multiple causal variants' to PLOS Genetics.

The manuscript was fully evaluated at the editorial level and by independent peer reviewers. The reviewers made positive comments but also raised substantial concerns about the current manuscript. Based on the reviews, we will not be able to accept this version of the manuscript, but we would be willing to review a much-revised version. We cannot, of course, promise publication at that time.

If you decide to revise the manuscript for further consideration at PLOS Genetics, please aim to resubmit within the next 60 days, unless it will take extra time to address the concerns of the reviewers, in which case we would appreciate an expected resubmission date by email to plosgenetics@plos.org.

[LINK]

We are sorry that we cannot be more positive about your manuscript at this stage. Please do not hesitate to contact us if you have any concerns or questions.

Yours sincerely,

Heather J Cordell

Associate Editor

PLOS Genetics

David Balding

Section Editor: Methods

PLOS Genetics

Reviewer's Responses to Questions

**Comments to the Authors:**

Reviewer #1: Please see attached Word doc.

Reviewer #2: Review of Wallace

Summary

This paper introduces an extension of the "coloc" method for colocalization

to deal with multiple causal variants in a region. This extension exploits a

recently-introduced method for fine mapping (SuSiE). The extension is

attractive in its simplicity, and simulations show it to perform better than some

alternative approaches. The paper also suggests a way to speed up computations

by pre-filtering out "non-significant" SNPs.

The key idea of combining SuSiE and coloc is nice, and I think that with

some improvements to the presentation will make a nice publishable contribution.

The idea of speeding up SuSiE by pre-filtering SNPs is also attractive from

a practical point of view, but it has some potential downsides that I feel

are not sufficiently emphasized and explored (even though the manuscript does end

with a statement that trimming might be not beneficial in general final mapping).

Specifically trimming out non-significant SNPs

could increase the potential for false positive identifications,

and indeed such a result has been previously reported in

https://www.biorxiv.org/content/10.1101/631390v3

(their Figure S7). It's not clear to me how, if at all, this is reflected in the results

shown here. Maybe it is simply the case that, as the paper suggests in the discussion,

that "Coloc benefits from comparing posterior probabilities across... two traits".

But the overall way that the manuscript deals with false positive (or indeed

false negative) identifications

is not clear. (Maybe methods are applied with some

knowledge of the true number of causal effects? It isn't clear to me.)

Since there are also other potential ways to speed up computation (see comments below)

I am not really convinced that the pre-filtering approach is really the way to go,

and would like to see at least a stronger assessment of the potential downsides.

Main Comments

1. The presentation of the method requires more details, including more precise

equations showing how quantities computed by SuSiE are used/combined. For

example you could introduce $\\alpha_{lj}$ for the matrix of posterior probabilities output by susie

and then give explicit expressions for the Bayes Factors being computed

($BF_{lj}$) in terms of $\\alpha_{lj}$. I'm not sure what $P_0$ is (is it something output by SuSiE?)

Is $\\pi=1/p$ where p is the number of SNPs in the region, or something else? How

do you set the maximum number of effects in SuSiE (L in the SuSiE paper)? Do you get SuSiE to

estimate the number of effects by estimating the prior variance, or do fix the prior variance?

If $L_g$ is the number of effects identified by SuSiE in the GWAS and $L_e$ the

number identified by SuSiE in the eQTL study, do you end up running coloc $L_g * L_e$ times?

(as suggested by "for every pair of regressions across traits" on p3).

How do you combine/summarise the results from all these different runs of coloc?

2. Presentation of colocalization results also needs more details. Can you say explicitly

what is an "AA" or "BB" comparison and an "AB-like signal"? From the description on p3 I

thought the simulations would include settings where there were 2 causal variants in each trait,

but no sharing. But Fig 3 seems to suggest

only a small portion of potential configurations of up to 2 signals in each trait are actually

included - is that right? (why?) And in Fig 3, what happens if SuSiE finds a signal in one trait

and not in the other - what comparison do you make? (Or do you force SuSiE to find the right

number of effects in each trait by fixing L to the true value? If so, is that cheating?)

Is the smaller height of the AA bar for susie_0 compared with other methods -- and indeed

the slightly smaller height of all bars -- something to be

concerned about? Are all methods equally applicable if (as is always the case) you do not know

the true number of causal signals in each trait?

3. Figure 1 compares only the PIPs at causal variants. Since in practice we don't know the

causal variants, one should also care about PIPs at non-causal variants. Is there a tendency

for SuSiE to inflate PIPs at non-causal variants when trimming?

4. It seems there are many potential ways to improve computation than

filtering out non-significant SNPs, and many of them may ultimately be better choices

(although filtering is obviously very simple to implement!) I don't think the discussion

in the paper really adequately reflects the options available or the many

issues involved.

Although I did not see it explicitly said anywhere, I believe the

paper is using the susie_rss function for applying SuSiE to summary data.

The details of this function are not included in the original SuSiE publication, but at time of writing

this function works by performing an initial eigendecomposition of the reference LD matrix R, which

makes it possible to convert the summary data into "transformed data" to which

regular SuSiE can be applied. This approach is appealing from a software engineering

point of view, but not necessarily the most efficient, computationally. The eigendecomposition

of R is quite expensive, being O(p^3) where p is the number of SNPs.

The subsequent application of SuSiE

to the transformed data is O(p^2) per iteration.

Thus if p is sufficiently large the eigendecomposition step will likely

dominate the susie_rss computation (and Figure 2 does indeed suggest computation maybe

increase something like p^3?)

One way to reduce computational complexity would therefore be to avoid the eigendecomposition

step, and we are currently actively exploring these in our development of susie_rss.

However, note that computing R itself is already

an O(np^2) operation, where $n$ is the number of samples in the reference sample used to compute R. So

if n is big then this computation (which is basically considered free

in this paper since R is precomputed) could be the dominant computational cost. Alternatively

if n<<p, --="" avoid="" entirely="" forming="" one="" perhaps="" r="" should="" then="">in the case n<</p,>

SVD of the reference genotypes (O(n^2p)) which will cheaper than forming R (O(np^2)) when n<<p.

In the future it seems quite likely that pre-computed R and eigen(R) could be made

available for some large panels, avoiding the need for each user to compute them. Once

these pre-computations are done there may no longer be any need to filter SNPs.

Other comments/details

- p3 although the number of potential models increases exponentially, SuSiE computation

does not increase exponentially.

- p4: "We labelled each comparisons considered...." I did not understand this sentence.

- p4: "... having strongest posterior support for H_4" - this should be H_3?

- p8: " this does apply to single trait" - missing *not*?

- In the second row-set of Figure 3, is the figure on the LHS

wrong? (The methods suggest colocalization but the figure shows no shared variant...)

- on p7 the r2 threshold is 0.8 but on p4 it is 0.5. Are there referring to different thresholds?

This review is signed: Matthew Stephens</p.

Reviewer #3: This is an interesting paper. The method is solid and implements M Stephen group's SUSIE method in the coloc framework with some simulation based comparisons with other methods (and "trimming" rather than shrinkage to help compute time). Expanding coloc to multiple variants is a useful advance to the field, and that is what PLoS Genetics Methods section papers are supposed to do.

I only have minor comments.

The formatting of figure 3 - the scenarios - seems to have gone slightly awry and needs to be fixed.

I suggest the discussion could be extended slightly - it is rather brief (although sufficient).

**Have all data underlying the figures and results presented in the manuscript been provided?**

Reviewer #1: Yes

Reviewer #2: Yes

Reviewer #3: Yes

PLOS authors have the option to publish the peer review history of their article (what does this mean?). If published, this will include your full peer review and any attached files.

Reviewer #1: No

Reviewer #2: **Yes: **Matthew Stephens

Reviewer #3: No

---

## [Decision Letter · Decision Letter 1]

2 Jul 2021

Dear Dr Wallace,

Thank you very much for submitting your Research Article entitled 'A more accurate method for colocalisation analysis allowing for multiple causal variants' to PLOS Genetics.

The manuscript was fully evaluated at the editorial level and by independent peer reviewers. The reviewers appreciated the attention to an important topic but identified some concerns that we ask you address in a revised manuscript

We therefore ask you to modify the manuscript according to the review recommendations. Your revisions should address the specific points made by each reviewer.

[LINK]

Yours sincerely,

Heather J Cordell

Associate Editor

PLOS Genetics

David Balding

Section Editor: Methods

PLOS Genetics

Reviewer's Responses to Questions

**Comments to the Authors:**

Reviewer #1: Please see attached Word doc.

Reviewer #2: Review of Wallace (revised)

Thank you for the responsiveness to issues raised in the previous review.

I have just two points to be addressed.

1. As noted, Susie is under active development, and

since v0.10.1 (March 16th 2021) the susie_rss function no longer performs

eigen-decomposition of R. This fact could be noted

in the discussion, and the version of susie

used to produce the results reported here should be reported.

2. I found Figure 1 hard to read. Most of the ink is not

very informative, and one has to read the actual numbers to

extract the information. Also change in total PIP is probably less relevant than

changes in individual PIPs (eg if all PIPs increase a very small amount,

the total change can be big, but it probably doesn't matter much.)

I think there should be better ways to convey the information.

Possibly a scatterplot of PIPs for each SNP, with vs without trimming,

might work - most of the points will presumably be near (0,0) but

any outliers should be immediately apparent?

**Have all data underlying the figures and results presented in the manuscript been provided?**

Reviewer #1: Yes

Reviewer #2: None

PLOS authors have the option to publish the peer review history of their article (what does this mean?). If published, this will include your full peer review and any attached files.

Reviewer #1: No

Reviewer #2: **Yes: **Matthew Stephens

---

## [Decision Letter · Decision Letter 2]

2 Sep 2021

Dear Chris,

Thank you very much for submitting your Research Article entitled 'A more accurate method for colocalisation analysis allowing for multiple causal variants' to PLOS Genetics.

The manuscript was fully evaluated at the editorial level and by one independent peer reviewer. The reviewer now recommends acceptance, but identified some minor concerns that we ask you address in a further revised version before we can formally accept your manuscript.

We therefore ask you to modify the manuscript according to the review recommendations. Your revisions should address the specific points made by each reviewer.

[LINK]

Yours sincerely,

Heather J Cordell

Associate Editor

PLOS Genetics

David Balding

Section Editor: Methods

PLOS Genetics

Reviewer's Responses to Questions

Reviewer #1: The manuscript is greatly improved since I last saw it. I have a few minor comments below, mostly to clarify some points that were not clear, but I trust the author will address them so I don’t need to see another revision of the manuscript before publication.

Minor comments:

I didn’t understand “ratio of effects” in this sentence: “Here, multiple causal variants are dealt with by requiring colocalisation across all causal variants in a region, and that the ratio of effects of each causal variant on the two traits is constant across variants.”

I didn’t follow this sentence, and how connected with the rest of the paragraph: “Thus, the user is presented with a list of tag SNPs per signal for each trait, and the matrix of pairwise posterior probabilities of H4 may be examined to infer which, if any, pairs of tags represent the same signal.”

“This situation is confusing, because the same signal in trait 1 appears to colocalise with different signals in trait 1.” Should this read “…signals in trait 2”?

I didn’t see where Fig. 2 is referred to in the text.

Why is there no green line in Fig. 3a? And why are there no green points show in Fig. 3a, d?

For completeness, I think Fig. 3 should also show trait 1, susie signal 2? (And similarly for Fig. S2.)

In the Fig. 3 caption you should also make clear what the truth is (I recognize that this is given in the text).

“S2 Fig shows an example where the stepwise approach is less able to correctly identify the separate signals.” By “stepwise approach” do you mean cond_abo?

The example in S2 Fig seems interesting and instructive—maybe it is worth putting in the main text? If I understand correctly, one difference is that susie iteratively improves the fit, whereas cond_abo does not iterate—it conditions B on A, then A on B, then stops. So perhaps one important improvement in susie is that it iteratively improves the fit until convergence? Perhaps this could explain why susie better identifies the signals in S2 Fig?

“However, when no credible sets can be detected with confidence by SuSiE, single-coloc may still be able to make some inference.” Do you know why susie fails in these cases? If there is an explanation, it would be helpful to add it here. I’m guessing that this failure occurs in cases where the support for association is not strong? On the surface the need for the “hybrid” method is a bit surprising, but there could very well be a good reason for it.

Under “Availability”, you might want to mention the susie vignette in the coloc package, which seems particularly helpful for those interested in applying the new susie-based coloc methods.

**Have all data underlying the figures and results presented in the manuscript been provided?**

Reviewer #1: Yes

PLOS authors have the option to publish the peer review history of their article (what does this mean?). If published, this will include your full peer review and any attached files.

Reviewer #1: No

---

## [Editor Report · Decision Letter 3]

12 Sep 2021

Dear Dr Wallace,

We are pleased to inform you that your manuscript entitled "A more accurate method for colocalisation analysis allowing for multiple causal variants" has been editorially accepted for publication in PLOS Genetics. Congratulations!

Yours sincerely,

Heather J Cordell

Associate Editor

PLOS Genetics

David Balding

Section Editor: Methods

PLOS Genetics

**Data Deposition**

http://datadryad.org/submit?journalID=pgenetics&manu=PGENETICS-D-21-00266R3

**Press Queries**

---

## [Editor Report · Acceptance letter]

23 Sep 2021

PGENETICS-D-21-00266R3 

A more accurate method for colocalisation analysis allowing for multiple causal variants 

Dear Dr Wallace, 

We are pleased to inform you that your manuscript entitled "A more accurate method for colocalisation analysis allowing for multiple causal variants" has been formally accepted for publication in PLOS Genetics! Your manuscript is now with our production department and you will be notified of the publication date in due course.

With kind regards,

Andrea Szabo

PLOS Genetics

On behalf of:
